# Clarification Processes of Orange Prickly Pear Juice (*Opuntia* spp.) by Microfiltration

**DOI:** 10.3390/membranes11050354

**Published:** 2021-05-12

**Authors:** Jaime A. Arboleda Mejia, Jorge Yáñez-Fernandez

**Affiliations:** Laboratorio de Biotecnología Alimentaria, Unidad Profesional Interdisciplinaria de Biotecnología, Instituto Politécnico Nacional, Av. Acueducto S/N Col. Barrio La Laguna, Ticoman, D.F. CP 07340, Mexico; albertomejia.it@gmail.com

**Keywords:** betalain content, clarification, microfiltration, transmembrane pressure (TMP)

## Abstract

In this study, fresh orange prickly pear juice (*Opuntia* spp.) was clarified by a cross-flow microfiltration (MF) process on a laboratory scale. The viability of the process—in terms of productivity (permeate flux of 77.80 L/h) and the rejection of selected membranes towards specific compounds—was analyzed. The quality of the clarified juice was also analyzed for total antioxidants (TEAC), betalains content (mg/100 g wet base), turbidity (NTU) and colorimetry parameters (L, a*, b*, Croma and H). The MF process permitted an excellent level of clarification, reducing the suspended solids and turbidity of the fresh juice. In the clarified juice, a decrease in total antioxidants (2.03 TEAC) and betalains content (4.54 mg/100 g wet basis) was observed as compared to the fresh juice. Furthermore, there were significant changes in color properties due to the effects of the L, a*, b*, C and h° values after removal of turbidity of the juice. The turbidity also decreased (from 164.33 to 0.37 NTU).

## 1. Introduction

Mexico is one of the largest producers of prickly pear around the world, with a production of 428,300 tons per year, or 44% of global production. Prickly pear shows great genetic variability, reflected in a wide range of colors (such as yellow, red, green, violet etc.) [1]. The *Opuntia ficus indica* is highly adaptive in different environmental conditions and can be planted in various ecological systems, making them an interesting agroindustrial resource [2]. *Opuntia ficus indica* plays several important roles, e.g., as a source of medicinal treatments and a traditional food. Prickly pear, also known as nopal fruit, is a native fruit of the Americas which grows in arid and semiarid regions. It has potential active nutrients and functional properties, including antioxidant and antiulcerogenic properties. These, in turn, have protective effects against peroxidation of high-density lipoproteins that are attributed to phenolic compounds, betalanic compounds (betaxanthins and betacyanins) and ascorbic acid [3,4,5]. Thus, prickly pear is an interesting object of study.

A notable characteristic of the nopal fruit is the orange red colorations, which are represented in two groups (betalains and betaxanthins), depending on their structural characteristics and light absorption properties and the pigment content of the vacuole (which replaces the anthocyanins in most families, including *Cactaceae*) [6,7].

In addition, the pigments of this fruit represent antioxidant properties, the most effective of which is ascorbic acid [8,9]. This fruit grows in different colors, depending on its betalains, covering a wide spectrum from white to purple with a pigment content of 66 to 1140 mg/kg of fruit pulp. They present a natural alternative to synthetic red dyes. In this way, the betalains of *Opuntia ficus indica* represent an interesting alternative and natural coloring agent, in addition to their antioxidant properties [8,10].

The low commercialization of *Opuntia ficus indica* is partly due to its physicochemical properties: high-water content, pH, content of soluble solids, and a considerable sugar content that make it susceptible to microbial attack during the postharvest [11].

Compared to traditional juice processing operations, membrane processes offer multiple advantages, e.g., smooth conditions, high separation and clarification capacity, low energy consumption, and easy scaling [12,13].

In this study, the effects of the microfiltration process (MF) on the physicochemical composition of orange prickly pear juice were investigated. Specifically, the effects of the process on the functional properties of the fruit were examined. In addition, different parameters of the microfiltration process (e.g., limiting transmembrane pressure (TMP_lim_) and rejection of compounds) were investigated.

## 2. Materials and Methods

### 2.1. Orange Pickly Juice (Opuntia spp.)

The orange prickly pear (*Opuntia ficus indica*) fruits used in this study were collected in the San Martin de las Pirámides area of the state of Mexico, Mexico, located northeast of the state of Mexico between latitude 19°37′05″ minimum and 19°46′20″ maximum; length 98°45′40″ minimum and 98°53′27″ maximum, with a height of 2300 msnm, at a distance of 40 kilometers from Mexico city.

### 2.2. Pretreatment of Orange Prickly Pear Juice

The fruits were washed with water and disinfected with sodium hypochlorite (Hycel, México) solution (1 mg/L). Subsequently, the juice was extracted using a ^®^TURMIX (Turmix, México) juice extractor. The juice extracted was centrifuged at 8200 RPM for 20 min in a Beckman Coulter centrifuge (model J2-MC, USA).

### 2.3. Microfiltration Unit and Filtration Experiments

The obtained supernatant was subjected to an MF process to remove suspended solids and high molecular weight compounds.

The MF process was carried out using a laboratory-scale membrane system unit equipped with a feed tank, a peristaltic pump, a cooling tank which utilized tap water, a pressure gauge and a pressure regulating valve, as shown in Figure 1.

The MF system was integrated by a hollow fiber membrane module (Amersham Biosciences Corp. Model CFP-1-E-4A, USA) of 0.1 Micron. Its specifications are shown in Table 1.

The MF processes were performed by recirculation configuration at a temperature of 20 ± 1 °C and a volumetric flux of 77.80 L/h. The different TMP that were used were 34, 69, 103, and 138 kPa—this was for the purpose of finding the TMP_lim_ of the system. The experimental tests were devoted to the characterization of the membrane for the clarification process and to the analysis of the physicochemical properties (e.g., the antioxidant capacity, betalain content, turbidity, brix degree, pH and colorimetry) of the juice in the process.

### 2.4. Clarification of Orange Prickly Juice by Batch Concentration Mode

The clarification of the orange prickly pear juice was performed via batch concentration mode (the permeate stream was collected by separate container and the retentate was recycled to the feed container). The MF process was achieved using 3675 mL of feed solution to collect a permeate solution of 3052 mL in order to reach a volume reduction factor (*VRF*) of 5.9. The *VRF* was defined as the ratio between the initial feed volume and the volume of the resulting retentate according to the following equation:(1)VRF=VfVr 
where *V*_*f*_ and *V*_*r*_ are the feed and retentate volumes, respectively.

### 2.5. Parameters of the Membrane

The behavior of the membrane was evaluated for productivity (permeate flux) and rejection of specific compounds.

The permeate flux was determined by measuring the collected permeate weight in a unit of time across the surface of the membrane using the following equation:(2)JP=WPA∗t
where *J*_*P*_ is the permeate flux (kg/m^2^·h), *W*_*P*_ is the permeate weight (kg), *t* is time (h) and *A* is the area of the membrane (m^2^).

The rejection of the selected membrane towards specific compounds was calculated with the following formula [13]:(3)R(%)=(1−CPCf)∗100
where *C**_P_* and *C**_f_* are the measurements of the concentration of specific compounds in the permeate and feed streams, respectively.

The fouling index (*FI*) of the membrane was determined according to the following equation:(4)$$FI(\%)=(1-(K_{p1}/K_{p0}))*100$$
where *K*_*p*0_ and *K*_*p*1_ are the pure water permeability before and after microfiltration, respectively.

After the clarification process, the membrane was subjected to an enzymatic cleaning treatment by means of MF, using ULTRASIL 67 (Ecolab, Minnesota) with a concentration of 0.5% (vol/vol) in water solution for 60 min at 55 °C [14].

The cleaning efficiency (*CE*) of the membrane was investigated as the flux recovery [15] according to the following formula:(5)$$CE(\%)=(K_{p2}/K_{p0})*100$$
where *K*_*p*2_ is the water permeability measured after the cleaning process.

### 2.6. Analytical Measurements

The permeate and retentate obtained during the microfiltration process were immediately frozen at −18 °C. The samples were analyzed for antioxidant activity, amount of betalains, colorimetry and turbidity.

The antioxidant activity was determined by the ABTS+ method, which is a free radical discoloration test in which the radical cation is generated by reaction with potassium persulfate before the addition of the antioxidant [16]. The spectrophotometric measurements were made using a spectrophotometer (lambda XLS Spectrometer). The ABTS+ (Sigma-Aldrich, St. Louis, MO, USA) was dissolved in 1 mL of sodium persulfate dissolved in water. It was completed to a volume of 5 mL with distilled water, then allowed to stand in the dark at room temperature for 12–16 h. Approximately 100 mL of 96% ethanol was added to the ABTS+ preparation to obtain an absorbance of 0.700 ± 0.020 at a wavelength of 734 nm. After the addition of 2 mL of ABTS+ to 0.020 mL of the sample, the absorbance was read at 1 m and 6 m. The value corresponding to 6 min was used to calculate antioxidant activity of the sample expressed in mg equivalent of Trolox. Each determination was made in triplicate and the result was expressed as the mean of the standard deviation of three samples.

The quantification of betalains was determined by the method described by Gandía et al. [17] and adapted by Viloria–Matos et al. [18]. The absorbance of the extracts (at 538 nm and 476 nm) was measured in a spectrophotometer (Lambda XLS Spectrometer). For the conversion of the absorbance units into concentration units, the following equation was used, in which it was expressed as grams of pigments (betacyanine or betaxanthin) per 100 g of sample on wet basis. The equation is the following:(6)mg pigment/100 g base sample=(A∗FD∗PM/E)
where *A* represents the absorbance at 476 nm for betaxanthin and 538 nm for betacyanin; *FD* is the dilution factor (in this case, 0.25); *PM* is the molecular weight of the pigment (betacinanines, 550.5 g/mol; betaxanthins, 339.3 g/mol); *E* represents the absorption coefficient or extension coefficient (betacyanine, 1120 L/mol·cm; betaxanthins 750 L/mol·cm) [19].

Turbidity was determined using an HI 93703 portable nephelometer (Hanna Intruments Inc., Woonsocket, RI, USA) before and after microfiltration.

Measurements of colorimetric parameters were determined as follows. First, 20 mL of orange prickly pear juice were placed in a petri dish. The colorimetric parameters—L*, *a** and *b**—were measured on the CIELAB scale using a colorimetric mark (^®^Konica Minolta colorimeter CR-10, Japan). Afterward, the values *h*° (angle hue) and *C* (chroma) were calculated using the following formulas:(7)$$h{}^{\circ}=\tan^{-1}(b^{*}/a^{*})$$
(8)$$C=(a^{*2}/b^{*2})*100$$

### 2.7. Statistical Analysis

Significant differences were evaluated by one way ANOVA (*p* ≤ 0.05). The analysis and Tukey’s HDS post hoc test were performed using Minitab^®^ 17.10 statistical software (Minitab, Ltd., Coventry, UK).

## 3. Results and Discussion

### 3.1. Effects of the Operating Parameters on the Permeate Flux

Several experiments were carried out with the microfiltration membrane module in order to analyze the behavior of the permeate flux. Figure 2 displays the evolutions of permeate fluxes as a function of operating time at different TMP. Additionally, Figure 2 shows that permeate flux decreased gradually with the operating time at different transmembrane pressures. During the process, two states were identified; in the first state, a dramatic drop of the permeate flux was noted in the first 12 min at all transmembrane pressures tested. Then, the permeate flux continued to drop slightly over 60 min of operation, until it reached values of 8.57, 24.85, 25.71, and 27.42 kg/m^2^·h for transmembrane pressures of 24, 69, 103, and 138 kPa, respectively.

The second state began after 70 min of operation, which was represented by a no-variation in the permeate flux as a function of the operation time. This state is well-known as the steady-state.

The gradual decrease of the permeate flux over operating time until it reached constancy can be attributed to the accumulation of juice components in the pores (membrane fouling) and membrane surface [20,21,22]. On the other hand, permeate flux behavior was directly proportional to the pressures applied in the system. These observations corroborated results by Galanakis et al. [23]. Those authors evaluated the performance of an ultrafiltration (UF) polysulfone membrane with a MWCO of 100 kDa in the recovery of phenol compounds from olive mill wastewater. They found that the UF membrane produced a steady-state of 76 L/m^2^·h at 200 kPa of TMP.

Other authors have also described the steady-state in different studies in the field of membranes; for instance, filtration of apple juice with a UF polysulfone membrane, with a MWCO of 100 kDa and 118 kPa of TMP [24], or treatment of kiwifruit juice with a UF polyvinylidenefluorid (PVDF) membrane, with a MWCO of 15 kDa and 90 kDa of TMP [25].

As expected, the filtering flux dropped to 78.58%. This was due to the accumulation of high molecular weight polysaccharides on the membrane surface and the phenomenon of polarization concentration.

Figure 3 shows the relationship between the permeate flux and the applied TMP. However, it was observed that as the TMP increased, a deviation of the linear flux pressure occurred and the flux became independent of the pressure.

Under these conditions, a limiting flux was achieved at a TMP value of 69 kPa (TMPlim = 69 kPa). Increases in pressure in the system did not correlate to a significant increase in the permeate flux. The presence of a limiting flux can be related to different fouling mechanisms and the polarization phenomenon of the concentration, represented by the feed extract that is directed by convection toward the membrane where the separation of suspended solids is carried out.

Figure 4 shows the time evolution of permeate flux and VRF in the clarification of the orange prickly pear juice with the MF membrane.

An agglomeration of the concentration of the extract was generated due to suspended solids that were rejected by the membrane. The permeate flux gradually decreased with the operating time, as VRF grew. This was due to different phenomena, including the concentration of polarization and fouling of the membrane [18,20]. In particular, the initial permeate flux of 34.29 L/m^2^·h decreased to about 9.14 L/m^2^·h—according to a final VRF value of 5.9.

### 3.2. Fouling Index and Cleaning Efficiency

Table 2 shows the hydraulic permeabilities, fouling index and cleaning efficiency determined in the MF membrane. The fouling of the membrane was caused by the accumulation of colloidal particles in the membrane, creating a resistance flow [26]. The fouling factor reduced permeate flux and productivity, increased feed pressure, decreased membrane life, and increased membrane maintenance and operating costs [27].

The fouling index of the MF membrane was determined on the basis of pure water permeability before and after the filtration process with the orange prickly pear juice.

The initial hydraulic permeability of the membrane was 44.72 L/m^2^·h kPa. After the MF process, it decreased by up to 30.44 L/m^2^·h kPa. The fouling index for the MF treatment was about 32%, as reported in Table 2.

Similar results (33.3%) were published by Gonçalves et al. [28] who investigated the optimization and removal of polysaccharides for MF in white wine. The fouling index in the current study was due to the main deficiency of polysulfone as a membrane material; the polymer is hydrophobic, which caused fouling of the membrane [29]. Similarly, it is known that polysaccharides (hydrophilic macromolecular compounds) can bind to less hydrophilic membrane surfaces through surface dehydration [30]. This renders the polysulfone membrane prone to fouling, due to the nature of its material and the matrix treated in the MF process.

Membrane fouling can. be influenced by different factors, including dissolved substances (proteins, polysaccharides, and polyphenols), polarization concentration, membrane characteristics (material, molecular weight cut-off, porosity, area charge, and membrane module), operating conditions, and electrostatic interactions [30,31].

Normally, an incomplete recovery in membrane permeability is attributed to the irreversible fouling factor, because polyphenols are capable of being absorbed by the membrane surface [32]. However, in the current study, an effective recovery of the initial permeability (91.2%) of the membrane was reported due to the efficiency of the enzymatic treatment used to remove the polysaccharides compounds from the surface of the membrane [33].

### 3.3. Influence of Microfiltration on the Physicochemical Properties of Juices

The physicochemical composition of feed, permeate, and retentate streams obtained during the MF of orange prickly pear juice is reported in Table 3.

The membrane process improved the physicochemical properties of the juice. For color measurements taken of the fresh juice and the clarified juice, the values were in the first quadrant of the CIELAB color system. The value of the hue angle changed from 26.83 to 31.55 in the clarified juice, indicating an increase in the degree of redness. The value of a* for the fresh juice and the clarified juice were positive, increasing from 16.7 to 21.56 which also indicates an increase in the color red.

The b* values increased from 8.36 to 13.2 and were positive, indicating a yellow color [34]. Regarding the parameters of luminosity and obscuration index, values of 28.6 for fresh juice and 31.4 for clarified juice were obtained, indicating increased clarity and decreased turbidity. Likewise, chromaticity values of 18.68 for fresh juice and 25.28 for clarified juice were obtained. The clarified juice presented a more vivid and striking color.

On the other hand, the orange prickly pear juice presented a weakly acidic medium with a pH value of 5.97. The prickly pear fruit is classified in the low-acid group (pH above 4.5) [35]. Similar values were found in the orange-yellow prickly pear (*Opuntia ficus indica*) with a pH value of 6.3 [8]. However, only a minimal change in the pH of the microfiltered juice (as compared to the fresh juice) was measured. This change can be attributed to the variability of measurement of the sample.

A greater increase in soluble solids was noted in the retentate (°Brix: 11.4) as opposed to the permeate (°Brix: 10.8). The retention of high molecular weight compounds by the membrane resulted in the presence of a high content of soluble solids, generating interference in the refractive index reading [36].

An increase in turbidity from 0.37 to a value greater than 1000 NTU was observed in the permeate and retentate, respectively. In addition, positive changes in terms of physical aspect were observed, as shown in Figure 5. Meanwhile, the permeate and retentate showed an increase in soluble solids from 10.8 to 11.4 °Brix, respectively. Small variations in the content of soluble solids resulted in large variations in turbidity values [37]. These types of filtration processes have been used as prior steps for other concentration processes, such as reverse osmosis [38]. Similar behavior was reported by Conidi et al. [12] during the treatment of artichokes using an ultrafiltration membrane with a molecular weight cut-off (MWCO) of 50 kDa (50,000 Da), in which an initial permeate flux of 19 kg/m^2^·h was observed, until a steady state of the permeate flux was reached at 10 kg/m^2^·h at VRF of 3.

Turbidity value is used for discrimination between juices of the same fruit species but with different concentrations, sizes and forms of particles. This parameter seems to be related to the volumetric concentration of solid particles, which are retained by a microfiltration membrane [36].

Table 4 shows the MF membrane’s rejection coefficients for different compounds. The reduction of compounds that contributed a volume of concentration to the orange prickly pear juice was notable, representing a turbidity recovery of 99.77%. Similar values with respect to turbidity have been seen in red wine lees microfiltration, with 77% recovery using a membrane with a pore size of 0.15 µm [39].

Regarding betalainic content: a value of 5.95 mg of pigment/100 g of sample was obtained for the fresh juice. For the clarified juice, the value was 4.54 mg of pigment/100 g of sample. The microfiltration membrane rejected 23.69% of betalainic compounds in the orange prickly pear. The rejection of these compounds was lower than that reported by Castro–Munoz et al. [38] using an ultrafiltration membrane module with molecular weight cut-off (MWCO) of 100 kDa (100,000 Da). Their study presented a retention rate of 5.52% for betalainic compounds.

The molecular weights of betaxanthins (339.3 g/mol) and betacyanins (550.5 g/mol) are much lower than the pore size of the membrane. Thus, variability was observed. On the other hand, these compounds tend to degrade due to various factors, e.g., temperature, nonoptimal pH, or the presence of oxygen [31].

The same behavior was presented in the study reported by Cassano et al. [34] in the clarification of *Opuntia ficus indica* (L.) Mill (yellow-orange) using an ultrafiltration membrane module with molecular weight cut-off (MWCO) of 10 kDa (10,000 Daltons). With respect to betalain quantities, they obtained results that showed 3003.7 mg of pigments/L in fresh juice and 2148.2 mg of pigment/L in clarified juice—a decrease of 43.52%.

There was a decrease of 45.32% in the antioxidant capacity of the clarified orange prickly pear juice, as compared to fresh juice. This could be explained by the degradation of the same compounds that provided antioxidant activity after exposure to different factors such as water, oxygen, and light [40].

This behavior was also reported by Cassano et al. [25] in their ultrafiltration of fresh kiwi juice using a polysulfone membrane module with molecular weight cut-off (MWCO) of 15 kDa (15,000 Daltones). Their study reported 17.6 mM Trolox in fresh juice and 16.2 mM Trolox in clarified juice, a decrease of 8% in antioxidant capacity.

Despite the minimal decreases observed in pigment and antioxidant capacity, the process of microfiltration by membrane is recommended, due to the gentle separation process, the conservation of bioactive compounds, the reduction of energy consumption, and the lower associated equipment costs [41,42,43].

After membrance microfiltration, the clarified product showed preservation of its bioactive compounds, high productivity, good physical characteristics and a reduction of suspended solids. Additionally, the present study highlighted the advantages of the membrane filtration process—high productivity, high selection, and the absence of extra phases [44].

## 4. Conclusions

The best TMP to clarify orange prickly pear juice was 69 kPa.

Values obtained for various parameters—luminosity, chromaticity, and darkening index—showed increased clarity and decreased turbidity. The increased chromaticity values for the clarified juice of orange prickly pear also led to a more vivid and striking color.

Decreased betalain content and reduced antioxidant capacity were observed during the MF process. This could be explained by the degradation of bioactive compounds that provide antioxidant activity due to different factors, such as exposure to oxygen and light. Despite the decrease in in pigment in the juice, the process of membrane microfiltration is recommended due to the gentle treatment during separation, the reduced damage to the product (due to the lack of heat treatments), the reduction of energy consumption, and the lower equipment costs.

The important properties observed, the desirable change in color, and the decrease in turbidity all represent possible benefits for commercial production.

## Figures and Tables

**Figure 1 membranes-11-00354-f001:**
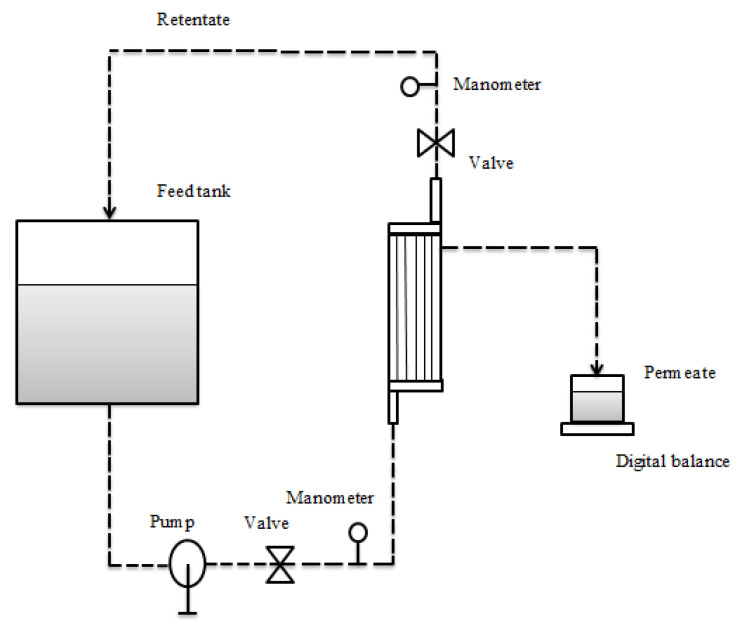
Schematic representation of the microfiltration process.

**Figure 2 membranes-11-00354-f002:**
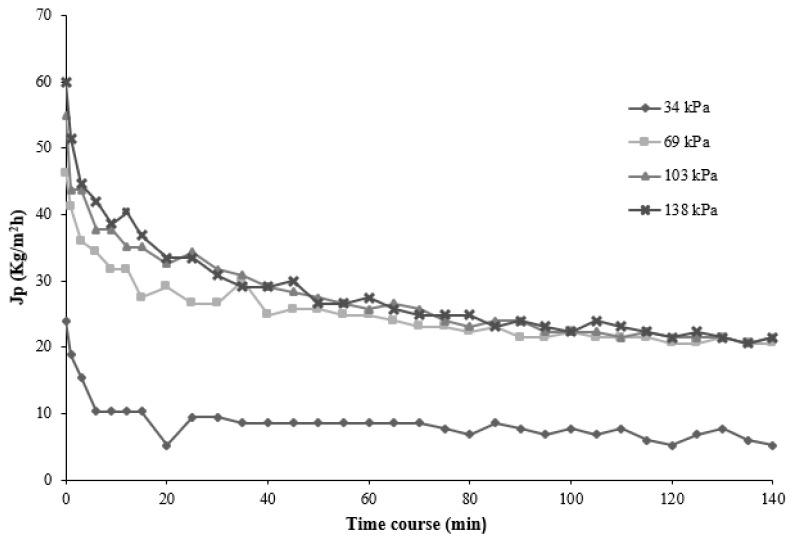
Permeate flux as a function of operating time at different transmembrane pressures (Operating conditions: T = 21 °C, Qf = 77.8 L/h).

**Figure 3 membranes-11-00354-f003:**
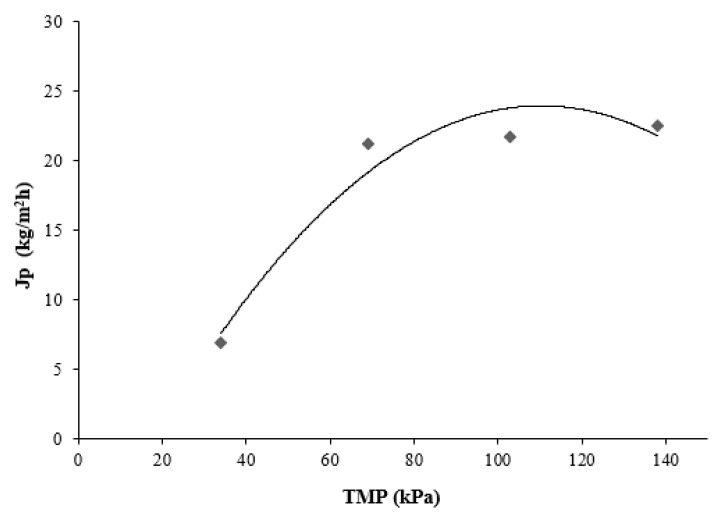
Permeate flux as a function of TMP. Conditions (T = 21 °C, Qf = 77.8 L/h).

**Figure 4 membranes-11-00354-f004:**
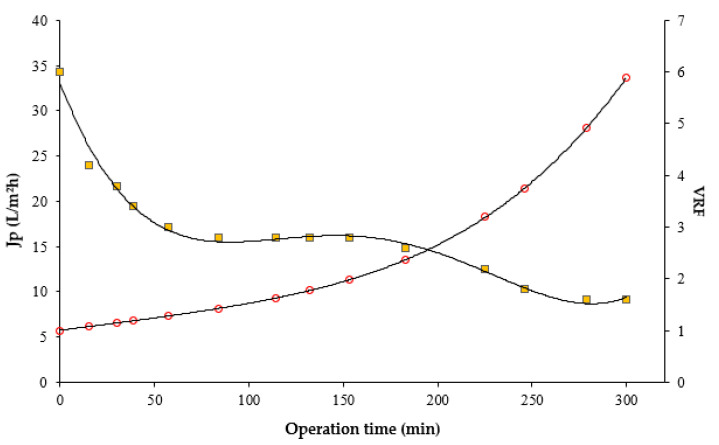
Operating time of the permeate flux and VRF (Operating conditions: PTM = 69 kPa, T = 21 °C, Qf = 77.8 L/h).

**Figure 5 membranes-11-00354-f005:**
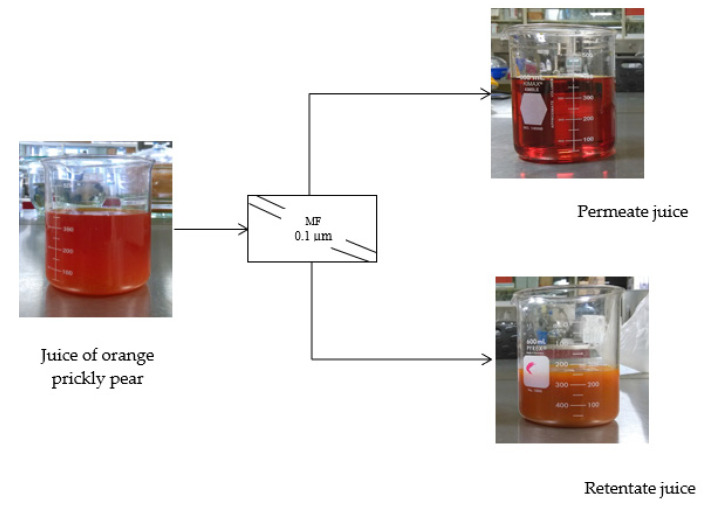
Schematic representation of the microfiltration process of the orange prickly pear juice.

**Table 1 membranes-11-00354-t001:** Operating specifications of the microfiltration membrane.

Membrane	MF
Manufacturer	Amersham Biosciences
Membrane type	CFP-1-E-4A
Nominal pore size (µm)	0.1
MWCO (Da)	1,000,000
Membrane surface area (cm^2^)	420
Membrane material	Polysulfone
Configuration	Hollow fiber
pH operating range	2 to 13
Temperature range (°C)	Up to 80 °C

**Table 2 membranes-11-00354-t002:** Hydraulic permeabilities, cleaning efficiency, and fouling index of the MF membrane on processing of orange pricky pear juice.

Membrane Parameters	
K_p0_ (L/m^2^·h kPa)	44.72
K_p1_ (L/m^2^·h kPa)	30.44
K_p2_ (L/m^2^·h kPa)	40.78
Fouling index (%)	31.93
Cleaning efficiency (%)	91.20

**Table 3 membranes-11-00354-t003:** Physicochemical properties of the juice before and after the clarification process.

Parameter	Feed	Permeate	Retentate
Antioxidant capacity (TEAC)	3.71 ± 0.71 ^a^	2.03 ± 0.13 ^b^	0.16 ± 0.01 ^c^
Content of betalains (mg/100 g wet base)	5.95 ± 0.05 ^a^	4.54 ± 0.01 ^b^	2.46 ± 0.01 ^c^
TSS (°Brix)	11.4 ± 0.23 ^a^	10.8 ± 0.07 ^b^	11.4 ± 0.12 ^a^
pH	5.97 ± 0.02 ^b^	6.72 ± 0.06 ^a^	5.04 ± 0.19 ^c^
Turbidity (NTU)	164.3 ± 12.4 ^b^	0.37 ± 0.10 ^c^	>1000 ± 0 ^a^
Colorimetry			
L	28.6 ± 1.78 ^b^	31.4 ± 0.61 ^a^	26.7 ± 0.78 ^b^
a*	16.7 ± 5.8 ^a^	21.5 ± 2.64 ^a^	5.3 ± 1.50 ^b^
b*	8.36 ± 2.16 ^b^	13.2 ± 0.92 ^a^	5.0 ± 0.90 ^c^
C	18.6 ± 6.2 ^a^	25.2 ± 2.7 ^a^	7.29 ± 1.69 ^b^
H	26.8 ± 2.02 ^b^	31.5 ± 1.40 ^b^	43.5 ± 3.74 ^a^

The values followed by the different superscript letters indicate a significant difference according to the Tukey’s HSD test (*p* ≤ 0.05).

**Table 4 membranes-11-00354-t004:** Rejection of the compounds by the microfiltration membrane.

Compounds	Rejection (%)
Antioxidant capacity	45.28
Content of betalains	23.69
TSS (°Brix)	5.26
Turbidity	99.77

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
