# Peer review of "Clarification Processes of Orange Prickly Pear Juice (Opuntia spp.) by Microfiltration"

_membranes, 2021, doi:10.3390/membranes11050354_

Round 1

Reviewer 1 Report

Manuscript ID: membranes-1191731Title: "Clarification processes of orange prickly pear juice (Opuntia spp.) by microfiltration" for Membranes.

Comments:

This paper is focused on the clarification of orange prickly pear juice by means of the microfiltration (MF) process with a commercial polysulfone membrane. The influence of the transmembrane pressure (TMP) has been investigated and the rejection parameters of compounds have been analyzed and compared to data literature.

The interest of this paper lies only on the applicative aspect and which shows a certain efficiency of the MF process. However for this kind application, the use of the MF process with a polysulfone membrane is not original. Fundamentally, this work is limited, therefore it would have been interesting to carry out a more exhaustive study by analyzing other influencing parameters, such as temperature (and for different pressures), the volumetric flux but also long operating times, treatment cycles (reuse of membranes over time), fouling, etc.

Also, for a complete analysis, it would have been useful to characterize the polysulfone membrane before and after use.

The interpretation of the results is light and little supported by experimental data that would support some hypotheses.

For a new improved version, I suggest that the authors also consider these following points:

  1. Equation (2) for the rejection parameter is not correct with regard to the values in Table 3. It shoul be ((Cf-Cp)/Cf) x100. Authors must clarify terms of rejection and retention. In the transport analyses, recovery and accumulation factors could be also considered.
  2. In table 2, the MWCO of the polysulfone membrane should be added. Indeed, authors compare the molecular weight of betaxanthins and betacyanins to the pore size but it will more convenient to use MWCO, even if 0.1µm is largely higher for compounds less than 500kDa.
  3. Figure 3, authors show that the TMPlim is reached for 10 psi. This is correct according to permeate fluxes in figure 2. However, the total operating time in figure 2 was 140 min. while in Figure 4 the time is longer (300 min) and a change of the permeate flux was observed. Is this change is observed for all tested pressures (5, 15 and 20 Psi)? What would have been the permeate flux at time longer than 300 min.?
  4. Mistakes or incorrect or incomplte sentences need to be corrected: p.6 line 181 “The membrane process improvement favorably…” (improves ?)p.7 line 210 “ using a ultrafiltration…(an ?)
  5. etc.
  6. p.7 line 200 “..was evident” (evidenced ?)
  7. p.4, line 139 “The one-way…were performed using…?”

Reviewer 2 Report

The article presented for review describes the research on the possibility of using microfiltration in the clarification process of orange prickly pear juice.

The prepared article requires minor corrections which I present below.

Chapter 2.3. (line 79): According to SI standards, the unit of pressure is not "psi".

Chapter 2.4. (line 96): The wrong formula for rejection was used. The correct formula is as follows:

R=[1-Cp/Cf] 100%

Chapter 3.1.

  • (lines 142-154): Please discuss the results in more detail.
  • (line 155): It is not true that: “Figure 3 shows the linear relationship between the permeate flux and the applied 155 TMP”.
  • (line 156): Misspelled TMP abbreviation.
  • (line 158): Why is the graph extrapolated to zero when the first working pressure is 5 psi?
  • (line159): The caption in the Figure 3 is incorrect as it concerns TMPlim.
  • (line 168): What does VRF stand for? There is no development of it in the text. If it is a volume reduction factor, give the formula as it was calculated.

Round 2

Reviewer 1 Report

Thank you.

Author Response

Thanks for your comments throughout the reviews. 

Reviewer 2 Report

Comments:

The paper has been improved in the revised version. However, there are still some changes and corrections to be made:

  • Line 114 p.3, Kp parameters correspond to permeation fluxes and not permeability coefficients (as unit of K is L/m²/h)
  • Line 108 p.3 Equation 3 shoud be 1-CP/Cf
  • line 124 p.4, again Kp2 corresponds to water permeation flux and not permeabiity.
  • Line 166 p.5 “were performed” instead of “was performed”. In the first version, it was correct but the sentence was incomplete.
  • Line 218 p.7, sentence incomplete
  • Line 253 p.8 “remove” instead of “removal”.

Author Response

Thanks for your comments throughout the reviews. 

Please, see here the response to the comments point-by-point. 

  • Line 114 p.3, Kp parameters correspond to permeation fluxes and not permeability coefficients (as unit of K is L/m²/h). 

Response: The hydraulic permeability units (kp0, kp1 and kp2) were modified (In the table 2, and lines 239, 240). 

The modification is about the addition of the pressure unit (kPa). The unit of hydraulic permeability adding the pressure unit is as follows: (L/m2 h kPa)

  • Line 108 p.3 Equation 3 shoud be 1-CP/Cf

Response: The order of the parameters were modified in the Equation 3

  • line 124 p.4, again Kp2 corresponds to water permeation flux and not permeabiity.

Response: as answered in the first comment. : The hydraulic permeability units (Kp0, Kp1 and Kp2) were modified (In the table 2, and lines 239, 240). 

  • Line 166 p.5 “were performed” instead of “was performed”. In the first version, it was correct but the sentence was incomplete.

Response: The entire sentence (from line 164 to 166) was removed to replace it with the new sentence (Lines from 164 to 166).

  • Line 218 p.7, sentence incomplete

Response: The entire sentence was replaced for a new sentence  (Lines from 224 to 226).

  • Line 253 p.8 “remove” instead of “removal”.

Response: Done